# A Field Efficacy Trial of Recombinant Porcine Circovirus Type 2d Vaccine in Three Herds

**DOI:** 10.3390/vaccines11091497

**Published:** 2023-09-16

**Authors:** Lanjeong Ju, Usharani Jayaramaiah, Min-A Lee, Young-Ju Jeong, Su-Hwa You, Hyang-Sim Lee, Bang-Hun Hyun, Nakhyung Lee, Seok-Jin Kang

**Affiliations:** 1Division of Viral Diseases, Animal and Plant Quarantine Agency, 177, Gimcheon-si 39660, Gyeongsangbuk-do, Republic of Korea; lanjeong@korea.kr (L.J.); eushavet85@gmail.com (U.J.); ma5147@korea.kr (M.-A.L.); ysh0108@korea.kr (S.-H.Y.); leehs76@korea.kr (H.-S.L.); hyunbh@korea.kr (B.-H.H.); 2Technology Institute, KBNP, Anyang-si 14059, Gyeonggi-do, Republic of Korea; yjjeong@kbnp.co.kr (Y.-J.J.); nhlee21@kbnp.co.kr (N.L.)

**Keywords:** recombinant porcine circovirus type 2d vaccine, porcine circovirus-associated disease, piglets, field trial, efficacy

## Abstract

This study aimed to evaluate the efficacy of a virus-like particle (VLP) vaccine containing the open reading frame 2 of porcine circovirus type 2d (PCV2d) in a farm environment where natural infections associated with porcine circovirus-associated disease are endemic. The vaccine trial was conducted on three farms (H, M, and Y) with a history of infections including porcine reproductive and respiratory syndrome virus (PRRSV), PCV, *Mycoplasma*, and *E. coli*. Farm H, as well as farms M and Y, experienced natural PCV2 infection between 4 and 8 weeks post-vaccination (wpv), and 8 and 12 wpv, respectively. Viremia levels of all farms were significantly (*p* < 0.05) lower in vaccinated piglets than the control group after natural infection. In all farms, serum immunoglobulin G levels peaked at 8 wpv in the vaccinated groups, surpassing those in the control groups. Furthermore, neutralizing antibody titers were significantly (*p* < 0.05) higher in the vaccinated groups than the control groups in farms H and Y (0–8 wpv). However, there were no significant differences between the vaccinated and control group in neutralizing antibody titers of farm M (0–20 wpv). In terms of body weight, vaccinated piglets from all three farms showed significantly increased average weights at 12 wpv compared to the control groups. In conclusion, our study revealed noteworthy differences in viremia and body weight gain between vaccinated and control animals on three farms. As a result, this field trial of PCV2d VLP vaccine was successful in protecting piglets from natural PCV2 infection.

## 1. Introduction

Porcine circovirus type 2 (PCV2) is a non-enveloped deoxyribonucleic acid (DNA) virus belonging to the family *Circoviridae*. It is the smallest known virus, measuring approximately 17 nm [1]. In addition, it is the causative agent of porcine circovirus-associated disease (PCVAD) in pigs [2]. In association with other pathogens, such as the porcine reproductive and respiratory syndrome virus (PRRSV), swine influenza virus, parvovirus, *Salmonella* spp., and *Mycoplasma,* PCV2 contributes to significant economic losses in the swine industry [3]. These losses are due to increased mortality of the pigs, reduced daily weight gain, and a delay in reaching market weight [4]. Once the PCV2 virus has infected the host, it is shed in high titers through natural secretions and excretions over prolonged periods, increasing the risk of transmission to other pigs and circulation within herds [5,6,7,8]. Regarding PCV2 genotypes, the most acceptable scheme allows the definition of eight genotypes (PCV2a to PCV2h) [9]. Of these, PCV2a, PCV2b, and PCV2d are the most commonly circulating genotypes worldwide [10,11,12]. In the 1990s, PCV2a was the dominant genotype; however, its prevalence in pigs was successfully reduced after the introduction of PCV2a-based vaccines. Recently, a global shift from PCV2a and PCV2b to PCV2d has been observed [13,14,15,16]. Currently, PCV2d is the predominant genotype in Korea, while PCV2a and PCV2b are less common [4,17].

A previous neutralization study showed that commercial vaccine antisera had limited neutralizing activity (approximately 30%) against PCV2d, while neutralizing antibodies (NAs) derived from PCV2d open reading frame 2 (ORF2) virus-like particles (VLP) showed broad efficacy against multiple PCV2 genotypes [18]. Thus, PCV2d vaccines could effectively counteract the circulating PCV2d genotype, preventing pathological damage by inducing broad cross-neutralization against field isolates and resulting in high levels of PCV2-specific IgG and NAs that reduce lesions in pigs. This study was conducted on a limited number of *Mycoplasma*-, PRRSV-, and PCV2-free piglets. Vaccine efficacy under controlled laboratory conditions may differ from that of field conditions, where uncontrolled factors such as disease prevalence, previous exposure, maternal antibodies, and environmental factors are involved. Therefore, the aim of this study was to evaluate the effectiveness of the PCV2d ORF2 (VLP) vaccine on a large scale under practical farming conditions. It involved a large number of piglets on several farms, considering various conditions such as coexistence with other pathogens and environmental factors. These results could provide practical insights into the use of the PCV2d ORF2 (VLP) vaccine and serve as a basis for further research to optimize its utilization in the swine industry. Moreover, this research could assist in the development of improved vaccination strategies to reduce the impact of PCVAD on pig health and productivity. Ultimately, both farmers and consumers will benefit from these advances.

## 2. Materials and Methods

### 2.1. Farms

Three farms were selected for the clinical field trial. Farm H housed 2500 animals, while farms M and Y housed 1500 animals each, with an average birth rate of 90% and a piglet mortality rate of 10–11%. The structure of each farm was similar to that of a conventional pig farm, with several pens separated by fences within a single barn. The pigs within these barns were at approximately the same stage of rearing and kept under uniform conditions, including room temperature, humidity, type of feed, and water supply. In addition, each pen was placed opposite each other to ensure the absence of bias in the observation and in rearing management. Infectious agents such as PRRSV, PCV, *A. pleuropneumoniae*, *M. hyopneumoniae*, and *E. coli* were prevalent in these farms. All farms had routine vaccination programs, as follows: Pigs in Farm H were vaccinated against foot-and-mouth disease (FMD), classical swine fever (CSF), and circovirus (commercial vaccine based on PCV2a); those in Farm M were immunized against FMD, CSF, and *M. hyopneumoniae*; and those in farm Y against *M. hyopneumoniae,* circovirus (commercial vaccine based on PCV2a), and FMD.

The piglets that participated in this trial were a triploid cross LYD (Landrace × Yorkshire × Duroc), the breed reared by most pig farmers in Korea. According to routine rearing protocols, male piglets are castrated at three days of age and weaned at three weeks of age. After weaning, all piglets were housed together according to the specific management programs of the farm. As the objective of this trial was to evaluate the efficacy of the vaccine in the context of actual farm conditions, piglets were randomly selected at three weeks of age. Each of the selected piglets was provided with a unique tag for identification. Piglets of either sex were assigned at three weeks of age to a specific test group as determined by the results of a randomization program (http://www.graphpad.com, accessed on 20 May 2021).

A total of 120 three-week-old piglets were selected from each farm and divided into two groups: a vaccine group of 90 animals evenly distributed over three different pens and a control group of 30 animals housed in another pen (Figure 1). All animals had ad libitum access to suitable feed and water and their pens were cleaned regularly according to farm procedures. The study was conducted under double-blind conditions as appropriate for a clinical trial. The group allocation of the pigs was not shared between the experiment designer and the husbandry staff. Observations of clinical signs were recorded for all pigs by experienced porcine veterinary practitioners, and their records were then compared between groups at the end of these trials to assess potential significant differences. The animals used in this trial were treated humanely, and the study followed the ethical guidelines established by the Institutional Animal Care and Use Committee (KBNP approval number; KBNP P-21-01). The experimental pigs were well managed under the guidance of the farm manager and veterinarian in accordance with the management program established for each farm. In particular, the observation and recording of clinical signs and the collection of samples were supervised by a professional farm veterinarian in accordance with the guidelines of the Animal Welfare Code of Korea.

### 2.2. Vaccination

The vaccine formulation contained 200 µg/mL of rPCV2d VLPs as a single dose and 10% carbomer (Lubrizol, Wickliffe, OH, USA). These VLPs were produced by expressing the ORF2 gene of the PCV2d genotype in a baculovirus expression system [18]. Each piglet in the vaccinated group received one dose intramuscularly, and the control animals were injected intramuscularly with 1× PBS. The piglets were vaccinated at three weeks of age (day one), and their health was monitored regularly for the next 20 weeks (until 23 weeks of age). Every four weeks, 15 piglets from the vaccine group and five piglets from the control group were randomly selected for measurement of body weight, blood viral load (viremia), serum IgG (using a commercial PCV2a-based enzyme-linked immunosorbent assay (ELISA) kit and an in-house PCV2d-based ELISA), and NA titer (Figure 1).

### 2.3. Clinical Signs and Body Weight

After vaccination, all pigs were observed daily for clinical signs such as rectal temperature, respiratory and digestive disorders, coughing, and lameness according to a routine vaccination and management program. Clinical signs were recorded from the beginning to the end of the trial using the farm veterinarian’s scoring system. Individuals with noticeable clinical signs received antibiotic treatment. In the event of significant distress or accidents resulting in obvious clinical signs, the decision to consider euthanasia as an option was made by the attending veterinarian. Body weight was recorded up to 12 weeks post-vaccination (wpv).

### 2.4. Quantification of PCV in Blood

Total nucleic acid was extracted from serum samples collected from the three different farms using a commercial kit (Dneasy Blood and Tissue kit, Qiagen, Hilden, Germany), and analyzed by quantitative RT-PCR using a LightCycler 480 II equipment (Roche Diagnostics, Mannhein, Germany). The genomic copy number of the PCV2 virus in serum samples was quantified using a PCR Kit (GeNet Bio, Daejeon, Republic of Korea) according to the manufacturer’s instructions. Subsequently, the melting curves of the amplified products were analyzed to verify the specificity of the assay, and those with a cycle threshold (Ct) < 35 cycles were considered positive.

### 2.5. Phylogenetic Analysis

The full length of the ORF2 gene was amplified from the viral genomic DNA using the HotStarTaq Plus Master Mix Kit and an ORF2-specific forward primer (5′-GGAATGGTACTCCTCAACTG-3′) and a reverse primer (5′-CTCGTCTTCGGAAGGATTAT-3′). The resulting PCR product (1061 bp) was purified using the QIAquick PCR Purification Kit (Qiagen), and its identity was confirmed by DNA nucleotide sequencing (Macrogen, Seoul, Republic of Korea). PCV2 ORF2-positive samples were analyzed at the CLC workbench, and their respective sequences were aligned to those of reference strains of PCV2a, PCV2b, PCV2c, PCV2d, and PCV2e. A phylogenetic tree was constructed from the aligned sequences using Neighbor-joining, Jukes–Cantor, and a bootstrap value of 1000. The sample sequences that formed branches with the reference strains are highlighted in the red dot boxes.

### 2.6. Neutralization Assay

The neutralization test was carried out using PCV2d, as described previously [19]. A volume of 100 µL of 200 TCID_50_ PCV2d was incubated with 100 µL of serially diluted sera from each animal for 1 h at 37 °C. This mixture was then added to 5 × 10^3^ PK15 cells seeded in four wells of a 96-well plate and incubated for 2 h at 37 °C. The cells were washed twice in 1× PBS, and fresh medium was added. Subsequently, the cells were fixed in 80% acetone at 5 dpi, and PK15 cells infected with PCV2 were stained as previously described [20]. The NA titers were calculated using the 90% virus neutralization test (VNT90), which is defined as the highest serum dilution that protects more than 90% of the cells from PCV2d infection.

### 2.7. Serology

Serum antibody levels against PCV2 were determined using a commercial ELISA kit (Median Diagnostics, Chuncheon, Republic of Korea) coated with PCV2a antigens according to the manufacturer’s instructions [18], as well as an in-house ELISA coated with PCV2d VLPs. Briefly, 96-well plates were coated with 100 ng of the PCV2d VLP antigens and blocked with a 1% blocking buffer (BSA in PBS). Serum samples from the study were diluted to 1:1600 and 100 µL was added to each well before being incubated at room temperature for 30 min. After three washes with 1× PBS containing 0.1% Triton X, secondary goat anti-pig IgG conjugated with HRP enzymes was added and incubated for a further 30 min at 37 °C. After three washes, 100 µL of substrate chromogen mixture (tetramethyl benzidine solution) was added to the wells, and the optical density of the color reaction was measured at 450 nm.

### 2.8. Statistical Analysis

The results of the replicate experiments are presented as means ± standard errors. To compare the groups, the statistical significance was determined using GraphPad Prism 7 software (GraphPad Software, Inc., San Diego, CA, USA). The statistical analysis included multiple *t*-test between the vaccine and the control group. A *p* < 0.05 was considered statistically significant.

## 3. Results

### 3.1. Serum IgG Levels

In farm H, the commercial ELISA-based serum IgG levels of vaccinated animals remained at baseline until 12 wpv and then gradually increased to 20 wpv. In contrast, the serum IgG levels of control animals were below the baseline until 8 wpv, then gradually increased and became significantly (*p* < 0.05) higher than those of the vaccinated group at 12 wpv (Figure 2A). The serum IgG levels in the vaccinated animals determined using an in-house ELISA kit coated with PCV2d VLPs increased from 4 wpv and peaked at 8 wpv, with an S/P ratio of approximately 2.5, which was significantly (*p* < 0.001) higher than that in the controls. Subsequently, the levels briefly decreased until 12 wpv and then gradually increased to almost 2.0 at 20 wpv. The S/P ratio in the control animals remained below the positive threshold up to 4 wpv, and subsequently increased rapidly, peaking at 12 wpv with an S/P ratio of 3.0, and remaining at this level up until 20 wpv. In addition, the IgG levels in the control animals were significantly (*p* < 0.01) higher than those in the vaccinated animals from 12 wpv to 20 wpv (Figure 2D, Appendix A).

In farm M, the serum IgG levels determined using the commercial kit were similar between the vaccinated and control groups throughout the experiment. At the beginning of the experiment, the serum IgG levels were high in both the control and vaccinated groups (Figure 2B). In the in-house ELISA experiment, the vaccinated animals had high serum IgG levels with an S/P ratio of 1.2, which gradually increased to a peak S/P ratio of approximately 2.5 at 8 wpv. Subsequently, the antibody levels fell briefly to 12 wpv for 4 weeks. The control animals in the experiment initially had high serum IgG levels (S/P ratio = 2), which dropped to their lowest at 8 wpv before gradually rising to 20 wpv. There was a statistically significant difference (*p* < 0.01) between the levels in the vaccinated and control groups at 8 wpv. However, the vaccinated and control animals had similar blood antibody levels from 12 wpv, and there was no statistically significant difference between the two groups (Figure 2E, Appendix A).

In farm Y, the serum IgG levels of the vaccine and control groups showed a similar pattern to that observed in animals from farm M when the commercial kit was used to determine serum IgG levels (Figure 2C). Overall, serum IgG levels were higher when measured with the in-house ELISA kit than with the commercial kit. The serum antibodies of the vaccinated animals measured using the in-house ELISA kit gradually increased until 8 wpv, slightly decreased from 8 wpv to 12 wpv, and then gradually increased until 20 wpv. The S/P ratio of the control animals was >0.4 at 4 wpv and gradually decreased to a negative level until 8 wpv. From 8 wpv, the serum IgG levels decreased slightly in vaccinated animals, while they increased in the controls, with no noticeable difference between them at 12 wpv. Up to this point, the serum IgG levels in the vaccinated animals were considerably higher than those in the controls. However, the serum IgG levels in the control animals increased significantly (*p* < 0.001) from 12 wpv compared with those in the vaccinated animals, peaking at 16 wpv and remaining constant until 20 wpv (Figure 2F, Appendix A).

### 3.2. Neutralizing Antibody

In farm H, the NA titer of the vaccinated animals increased at 4 wpv (111 ± 81) and remained nearly constant throughout the study up to 20 wpv (268 ± 244). This titer was significantly (*p* < 0.05) higher than in the control animals until 8 wpv. However, the NA titer in the control animals increased rapidly at 12 wpv (794 ± 781) and peaked at 20 wpv (2048 ± 0.0). The NA titers of the control group were significantly (*p* < 0.01) higher than those of the vaccinated animals from 12 to 20 wpv (Figure 3A, Appendix A).

In farm M, the NA titers were weakly detectable on day 1 in both the vaccinated (11 ± 6.6) and control (9.6 ± 3.6) animals. Afterward, the NA titers increased from 4 to 20 wpv with no significant difference between the groups, with the sole exception of 16 wpv (Figure 3B, Appendix A).

In farm Y, the NA increased gradually from 4 wpv (55 ± 67) in the vaccinated animals, reaching a peak titer at 20 wpv (450 ± 508). In contrast, the titer of the control animals rapidly increased from 12 (211 ± 454) to 20 wpv (1843 ± 458); however, it was detected at a low level until 8 wpv (below 18 ± 13) (Figure 3C, Appendix A).

### 3.3. Viremia

In farm H, the serum viral load of the vaccinated animals remained at baseline throughout the experiment, which was significantly (*p* < 0.001) lower than that of the control animals from 8 wpv onwards. At the beginning of the experiment, the viral load of the control animals was the same as the baseline, while the Ct value decreased from 8 to 12 wpv and subsequently increased slightly (Figure 4A, Appendix A).

In farm M, the viral load of the vaccinated animals remained close to baseline and increased slightly from 12 wpv, but was significantly (*p* < 0.05) lower than that of the control animals. In the control animals, the Ct values decreased from 12 to 16 wpv, and subsequently increased to 20 wpv. Notably, the Ct values in the vaccinated animals were significantly (*p* < 0.05) higher than in the controls from 12 to 20 wpv (Figure 4B, Appendix A).

In farm Y, a similar viral load to that of farm M was observed, with the viral load in both the vaccinated and control animals decreasing from 12 wpv. However, the Ct values in the vaccinated animals were significantly (*p* < 0.001) higher than in the controls from 16 to 20 wpv (Figure 4C, Appendix A).

### 3.4. Phylogenetic Analysis

PCV2 samples isolated from each farm were sequenced and phylogenetically analyzed against the reference sequences of PCV2a, PCV2b, PCV2c, PCV2d, and PCV2e. In farms H and Y, the samples branched with the PCV2d reference strain, indicating that these isolates were similar to PCV2d. In farm M, some of the isolates diverged with PCV2d and others diverged with PCV2b (Figure 5B). However, PCV2d was prevalent in all the farms (H, M, and Y), whereas PCV2b was only detected in farm M alongside PCV2d isolates (Figure 5).

### 3.5. Body Weight

The body weights of the vaccinated (n = 10) and control animals (n = 10) were compared from day 1 to 12 wpv (Appendix A). On average, vaccinated pigs gained 3.3, 4.0, and 3.6 kg more than control pigs at 12 wpv on farms H (Figure 6A), M (Figure 6B), and Y (Figure 6C), respectively. Furthermore, throughout the 12-week study, the body weight of the vaccinated pigs was significantly (*p* < 0.05) higher than that of the control group on all farms.

## 4. Discussion

Swine farms in the Republic of Korea suffer from PCV2 infection, which is characterized by multiple genotypes and ongoing genotype shifts. Recent surveys have shown that the majority of swine farms are infected with PCV2d. To date, vaccination is the primary method of disease prevention, and the vaccines used in Korean farms contain PCV2a antigen. However, the PCV2a NAs have limited efficacy against the circulating PCV2d strain [18,20]. Despite the scarcity of PCV2a isolates in Korean swine farms, PCV2d infections continue to cause substantial economic losses. In our previous study, we demonstrated that PCV2d VLPs have cross-neutralization activity against different PCV2 genotypes and are effective in protecting pigs against PCV infection at the laboratory level [18]. Here, we present the efficacy result of the PCV2d VLP vaccine in a large number of animals from three different farm settings in the Republic of Korea.

In this study, the commercial ELISA kit, which is based on PCV2a antigens, could not detect seroconversion resulting from the PCV2d vaccine injected in animals from farm H, leading to a consistently low S/P ratio. In contrast, the in-house PCV2d-based ELISA successfully detected seroconversion. This difference can be attributed to alterations in antigen compatibility and suggests that the vaccine-induced anti-PCV2d IgG showed stronger binding to the in-house PCV2d ELISA antigen than to the PCV2a antigen used in the commercial ELISA kit [18]. Similarly, the commercial ELISA kit failed to detect seroconversion in farms M and Y, with serum IgG continuing to decrease up to 8 wpv. However, a rapid increase in serum IgG levels was observed after vaccination when the in-house ELISA kit was used to measure IgG levels instead. Therefore, it is essential to consider the use of an appropriate IgG ELISA kit to accurately assess the immunogenicity of PCV2d vaccines and evaluate natural infections.

In Korea, PCV2 infection has caused significant economic damage to pig farms in the past due to the unavailability of a PCV2 vaccine. Since the release of the PCV2 vaccine in 2006, most farms have adopted its use, resulting in a PCV2 antibody-positive status in almost all piglets. In this study, piglets from farms M and Y showed high titers at the beginning of the experiment, as determined by both the commercial and the in-house ELISA kits. These high titers could be attributed to maternally derived antibodies (MDAs) acquired from sows vaccinated with the PCV2a-based vaccine or from natural infection. It is important to note that the commercial ELISA kit is not effective in the detection of antibodies against the PCV2d genotype. We found that the levels of MDAs, which were higher on farm M than on farm Y, had decreased by 8 wpv and 4 wpv, respectively. The decline in MDA levels typically occurs over a period of 2–15 weeks, depending on the initial antibody concentration. The impact of MDAs on vaccine efficacy remains controversial. Some authors have stated that it is preferable to establish the sow’s infection or vaccination history before vaccinating piglets at 3–4 weeks of age, as MDAs may interfere with the formation of vaccine-specific antibodies and reduce vaccine efficacy [21,22,23,24]. However, in our study, both IgG and NA antibodies increased in the experimental animals after vaccination, indicating that MDAs had no effect on vaccine efficacy, in agreement with what was reported by Um et al. [25] and Kim and Hanh [26].

In our study, PCV2d VLP vaccination induced antibody responses in all the farms, with NAs playing a crucial role in protecting the animals from infection. Over the 20-week duration of the study, the vaccinated groups consistently maintained viral loads at baseline levels, and antibodies from the vaccinated animals effectively prevented PCV2 infections. In contrast, the antibody titers of the control animals increased from 4 to 12 wpv; however, these antibodies proved insufficient to protect the animals from viremia. In particular, the animals from farm H lacked maternal or previous antibodies, making the control animals more susceptible to natural infection than those from farms Y and M. Nevertheless, the viral load of the vaccinated animals on farm H remained consistently low throughout the study, whereas farms M and Y exhibited slightly higher viral loads compared to farm H. Significantly, natural infection was observed in the control animals on all farms starting at 4 wpv. This phenomenon became evident on farm H at 4 wpv, and on farms Y and M at 8 wpv. This was substantiated by an increase in viremia and serum IgG antibody level in the corresponding farms during the specific periods. Notably, only the PCV2d genotype was detected on farms H and Y (as shown in the phylogenetic tree, Figure 5), whereas infections with both the PCV2b and PCV2d genotypes were observed on farm M. In this context, the PCV2d VLP vaccine effectively provided protection against the PCV2d genotype in all three farms, as well as against the PCV2b genotype on farm M.

Infectious diseases play a significant role in causing poor growth performance in farm animals. Slower growth rates lead to an extended time to reach market weight, thereby incurring additional production costs that burden the farm economy [27]. Furthermore, the PCV2 titer in the blood directly correlates with the average daily weight gain (ADWG). Also, a higher viral load corresponds to a lower ADWG, thereby prolonging the time required for animals to attain a marketable weight [28]. In the three farms investigated in our study, the vaccinated animals showed a significantly higher body weight compared to control animals. Moreover, the administration of a single dose of the recombinant PCV2d VLP vaccine effectively reduced the viral load resulting from natural infection. Vaccination also contributed to increased body weight, irrespective of the presence of maternal antibodies.

Assessing the efficacy of the PCV2 vaccine in on-farm trials poses challenges due to the fact that single circovirus infection in piglets typically does not result in obvious clinical signs. Therefore, gauging vaccine efficacy usually relies on two variables: notable differences in the occurrence or concentration of viremia after a decrease in MDA levels, and in ADWG level over the same period. Thus, vaccine success is indicated by either complete virus elimination or a reduction in viremia, coupled with a significant increase in the ADWG of vaccinated compared to unvaccinated pigs (Figure 4 and Figure 6). The negligible histopathological variation observed between the two groups is likely a result of insignificant histological difference caused solely by the circulating circovirus and ongoing non-clinical effects of other pathogens. With regard to circulating antibodies, the typical pattern of antibody levels is consistent for both vaccinated and non-vaccinated groups in virus-free farms. However, assessing vaccine efficacy solely through ELISA or changes in NAs becomes complex in farms where different circoviruses are circulating. Notably, in such scenarios, a clear pattern emerges: there is a reversal in antibody levels during the rearing phage, indicating the suppression of viremia by vaccine-induced antibodies. This pattern of antibody reversal has been repeatedly observed in previous studies [29,30], supporting our findings from the farm trials. Specifically, concerning viremia, a significant difference was evident between vaccinated and non-vaccinated pigs during the fattening stage, resulting from the suppression of early circovirus infection in young piglets. In summary, in light of the reversal of antibody levels, the reduction in viremia, and the increase in ADWG, we are confident that our results convincingly demonstrate the efficacy of the vaccine within a genuine farm environment.

## 5. Conclusions

Irrespective of the presence of MDAs, the PCV2d VLP vaccine demonstrated its efficacy in protecting animals from natural infection up to 20 wpv, reducing their viral load and increasing ADWG. These findings indicate that the vaccine has the potential to effectively control PCV2d infection in swine herds in Korea, as well as in other regions affected by the circulation of the PCV2d genotype. In addition, these results are particularly relevant to pig farmers and practitioners, as they provide insights into the outcomes that can be anticipated from using the new vaccine under real farm conditions, which are influenced by various factors. Therefore, these results provide practical data, rather than purely scientific, that would ultimately shed light on why the vaccine should be used to contain the predominant genotypes currently circulating on pig farms.

## Figures and Tables

**Figure 1 vaccines-11-01497-f001:**
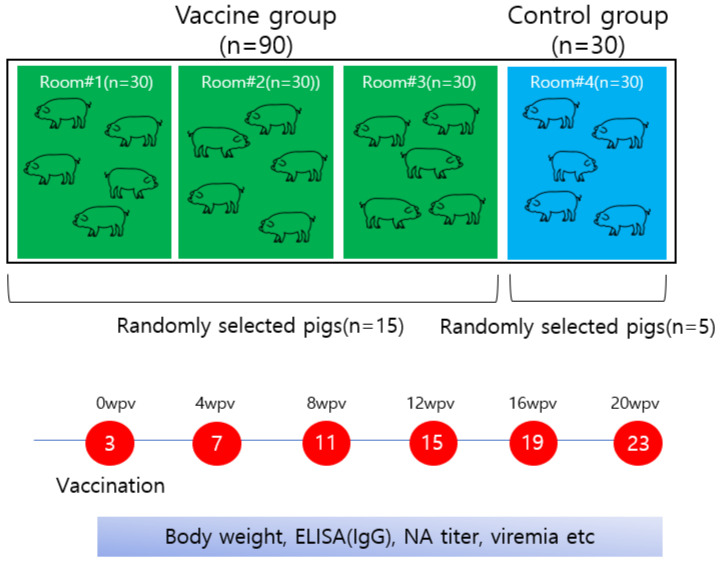
Diagram of the animal study carried out on three swine farms. wpv, weeks post-vaccination; ELISA, enzyme-linked immunosorbent assay; IgG, immunoglobulin G; NA, neutralizing antibody.

**Figure 2 vaccines-11-01497-f002:**
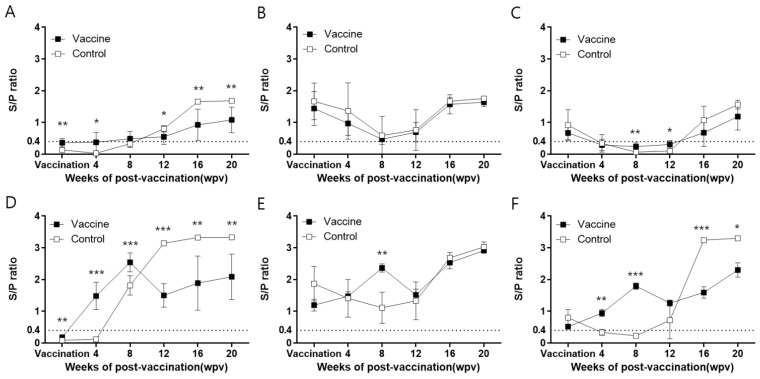
Evaluation of serum antibodies (IgG) in the three farms. The S/P ratio of all sera was measured using a commercial ELISA kit coated with PCV2a antigens (farm H (**A**), farm M (**B**), and farm Y (**C**)) and an in-house ELISA kit coated with PCV2d VLP (farm H (**D**), farm M (**E**), and farm Y (**F**)). All data are expressed as mean ± SE. *, **, and *** indicate significant differences (*p* < 0.05, *p* < 0.01, and *p* < 0.001, respectively) between groups by *t*-test (Holm–Sidak method). S/P, sample to positive; PCV2, porcine circovirus type 2; VLP, virus-like particle.

**Figure 3 vaccines-11-01497-f003:**
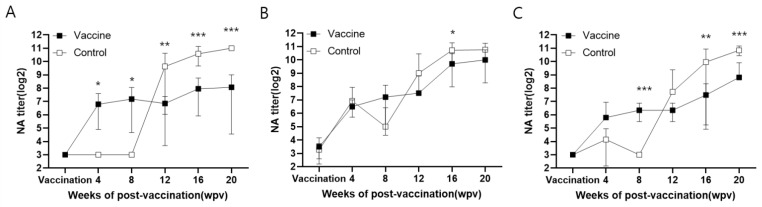
Evaluation of the neutralizing antibody (NA) titers. All sera from farm H (**A**), M (**B**), and Y (**C**) were neutralized with PCV2d isolate (QIA244) and the NA titers were calculated using VNT90. NA titers (log2) are expressed as mean ± SE. *, **, and *** indicate significant differences (*p* < 0.05, *p* < 0.01, and *p* < 0.001, respectively) between groups by *t*-test (Holm–Sidak method).

**Figure 4 vaccines-11-01497-f004:**
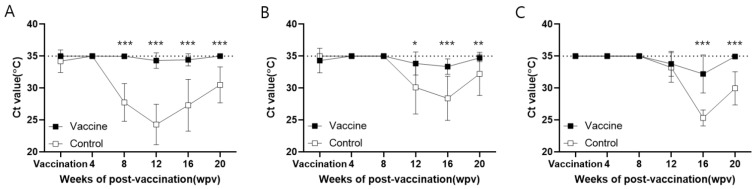
Evaluation of the viral load in the blood. Detection of the PCV2 genome is expressed as the mean ± SE Ct values (°C) of the vaccine (n = 15) and control groups (n = 5) in farms H (**A**), M (**B**), and Y (**C**). *, **, and *** indicate significant differences (*p* < 0.05, *p* < 0.01, and *p* < 0.001, respectively) between groups by *t*-test (Holm–Sidak method). Dashed lines at 35 °C represent a negative Ct value for PCV2 antigen.

**Figure 5 vaccines-11-01497-f005:**
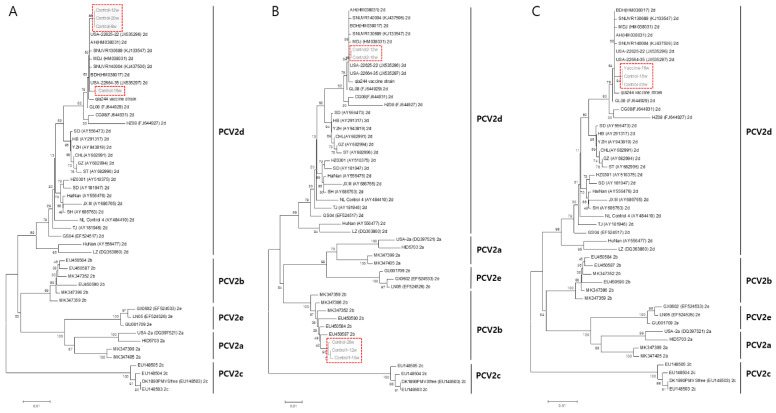
Phylogenetic analysis of PCV2 isolates in farms H (**A**), M (**B**), and Y (**C**). The red dot boxes indicate the PCV2 isolates from each farm.

**Figure 6 vaccines-11-01497-f006:**
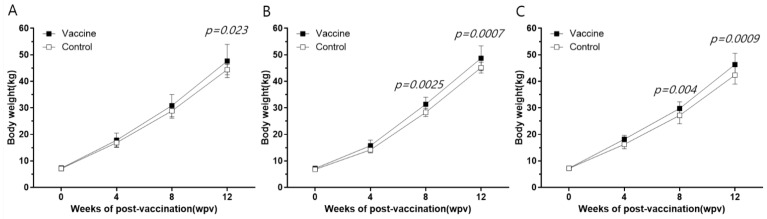
Comparison of body weight between the vaccine and control groups in farms H (**A**), M (**B**), and Y (**C**). All data are expressed as mean ± SE. Statistically significant difference between groups by *t*-test (Holm–Sidak method) is represented as *p*-value.

## Data Availability

The datasets generated or analyzed during this study are available form the corresponding author on reasonable request.

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
