# Peer review of "A Field Efficacy Trial of Recombinant Porcine Circovirus Type 2d Vaccine in Three Herds"

_vaccines, 2023, doi:10.3390/vaccines11091497_

Round 1

Reviewer 1 Report

Thank you for submitting this paper. Vaccines states that all submissions adhere to ARRIVE Guidelines for reporting the use of animals in experiments. As such, this paper falls short of these at this time. Please rewrite adhering to this guidelines and my suggestions, which are below.

-There is no justification of animal numbers included. Why is the control group 1/3 of the size of the treatment group? How did you chose the size of the experimental groups? What is your experimental unit-is it individual pigs or the rooms in which they were kept? How did you account for room differences?

-what breed of pig, what sex, how were they selected and were there any exclusion criteria-if so please state them

line 94; what method of randomisation was employed? If haphazard selection was used this must be stated rather than randomisation and this goes for selection of animals also.

-what level of blinding was used for the testing?

-clinical signs post vaccination; did you use scoring, were humane endpoints discussed, would animals have been given any treatment

-define how the animals were treated humanely, what mitigations were in place eg were the handlers experienced in pig handling and taking samples

-results; descriptive statistics for each Experimental group should be included

-will the raw data be made available

Author Response

Thank you very much for giving us a chance to revise our manuscript entitled ‘A Field efficacy Trial of Recombinant Porcine Circovirus Type 2d Vaccine in Three Herds’ by Ju et al. All comments made by the reviewers were quite valid and we sincerely appreciate for reviewers’ endeavor to improve the quality of our manuscript. We considered all suggestions of the reviewers for revising the original version of manuscript. We prepared a point-by-point report to explain our revision in details and also annexed a highlighted copy of the revised manuscript. However, we are still afraid of remaining of any mistake, and misunderstanding and neglecting of reviewers’ comments. We of course are ready for further revising our manuscript as the reviewers’ request. 

Thank you for submitting this paper. Vaccines states that all submissions adhere to ARRIVE Guidelines for reporting the use of animals in experiments. As such, this paper falls short of these at this time. Please rewrite adhering to this guidelines and my suggestions, which are below. -There is no justification of animal numbers included. Why is the control group 1/3 of the size of the treatment group? How did you chose the size of the experimental groups? What is your experimental unit-is it individual pigs or the rooms in which they were kept? How did you account for room differences?

To evaluate vaccine efficacy, we designed an experiment with 90 pigs in the vaccinated group and 30 pigs in the control group on each farm. For the experimental design, we focused on evaluating the protective efficacy of the vaccine due to funding considerations. Although the control group was one-third of the vaccinated group, we determined that 30 pigs was sufficient to assess statistical significance. Each animal was housed with 15 to 30 piglets in each pen, and the number of piglets in each pen was controlled and maintained according to the farm’s management program.The structure of each pig farm tested was similar to that of a typical conventional farm, with several rooms separated by fences within a single building(barn). The pigs within these barns were, as usual, at approximately the same stage of rearing. Consequently, pigs in all test groups were studied on identical farms and under uniform conditions. These included room temperature, humidity, type of feed, and water supply. In addition, each pen used in the test was placed opposite each other to ensure unbiased observation and impartiality of rearing management. We described in Materials and Methods (line74-79).

 -what breed of pig, what sex, how were they selected and were there any exclusion criteria-if so please state them

The piglets used in this trial were a triploid cross LYD(Landrace X Yorkshire X Duroc), which is the type used by most pig producers in Korea.According to the general routine of pig producers, male piglets are castrated at 3 days of age and weaned at 3 weeks of age. After weaning, all piglets were integrated with the male and barrow ones according to the specific management programs of the farm. As the objective of this trial is to evaluate the efficacy of the vaccine in the context of actual farm conditions, piglets were randomly selected at three weeks of age. Each of these selected piglets was then given a unique tag for identification purposes. Based on the pre-attached tags, each piglets was assigned to a specific test group as determined by the results of the randomized program. We described in Materials and Methods (line86-95).

 line 94; what method of randomisation was employed? If haphazard selection was used this must be stated rather than randomisation and this goes for selection of animals also.

The required number of 3-week-old piglets, just before weaning, were selected without regard to sex. All piglets were tagged and individually identified, and then randomly assigned to the test groups as follows.Using a randomization program, all piglet numbers were assigned to two test groups, and the required number of piglets for each test group was determined in three replicates.  Piglets with tags corresponding to each test group, as determined by through randomization, were then assigned to their respective test groups, and recorded. These records were accessible only to the experimental designer (investigator) and were not shared with the executor (worker, practitioner) responsible for supervising the trial.

 -what level of blinding was used for the testing?

As a clinical trial, it was conducted as a double-blind, two-group trial as follows. As described above for the randomization of piglets, the randomized number of pigs in each group was not shared between the experimental designer(investigator) and the executor (involved workers and pig practitioners), so that the performer could not identify which animal belonged to which group. Observations of clinical signs were recorded for all pigs by experienced pig practitioners, and these records were then allocated into each group at the end of these trials to assess significance between groups. In the case of blood sampling, the number of the pigs to be sampled was determined prior to sampling using a randomization program, and the pig practitioner collected blood from the pigs corresponding to the determined number in accordance with IACUC guideline.We described in Materials and Methods (line100-104).

 -clinical signs post vaccination; did you use scoring, were humane endpoints discussed, would animals have been given any treatment

Throughout the trial, all three farms maintained a routine vaccination and management program.The clinical signs of the pigs in this trial were recorded from the beginning to the end of this trial using the farm veterinarian’s scoring system. Emphasis was placed on scoring clinical signs specific to the circovirus.Individuals showing notable clinical sings were treated with therapeutic antibiotics.  In the event of significant distress or accidents resulting in obvious clinical signs, the decision to consider euthanasia as an option was made by the pig practitioners (experienced veterinarians). We described in Materials and Methods (line130-135).

 -define how the animals were treated humanely, what mitigations were in place eg were the handlers experienced in pig handling and taking samples

The experimental pigs were well managed during the trial under the guidance of the farm manager and the pig practitioner (managing veterinarian), in accordance with the management program established for each farm. In particular, the observation and recording of clinical signs and the collection of samples were supervised by a professional farm veterinarian throughout the trial (line107-111).

 -results; descriptive statistics for each Experimental group should be included

Described Statistics in each figures.

 -will the raw data be made available

Raw data will be available in supplementary data 1, 2, 3 & 4.

Thank you.
Sincerely yours,

Seok-Jin Kang

Reviewer 2 Report

Much as I appreciate the authors effort in the study, It was not clear to me what the authors try to achieve and I think that the design of the study is not appropriate. If the authors aim at evaluating the vaccine effect at field level, then the work performed is not satisfactory and further analysis should be performed (i.e. other methods to detect virus infectivity and pathological effect. My main concerns are that:

1- It seems that most of the responses detected are due to the field infection and the effect of vaccination is minimal.

2-The different farms received different vaccination regimes.

3- The authors have not tested for maternal antibodies which may affect the results.

4- The authors used PCR as an indication of infection instead of the virus isolation. Neither the sequence of the primers used nor the target product was provided which important to evaluate the study.

Some typos and wrong phrases were used.

Author Response

Thank you very much for giving us a chance to revise our manuscript entitled ‘A Field efficacy Trial of Recombinant Porcine Circovirus Type 2d Vaccine in Three Herds’ by Ju et al. All comments made by the reviewers were quite valid and we sincerely appreciate for reviewers’ endeavor to improve the quality of our manuscript. We considered all suggestions of the reviewers for revising the original version of manuscript. We prepared a point-by-point report to explain our revision in details and also annexed a highlighted copy of the revised manuscript. However, we are still afraid of remaining of any mistake, and misunderstanding and neglecting of reviewers’ comments. We of course are ready for further revising our manuscript as the reviewers’ request. 

Much as I appreciate the authors effort in the study, It was not clear to me what the authors try to achieve and I think that the design of the study is not appropriate. If the authors aim at evaluating the vaccine effect at field level, then the work performed is not satisfactory and further analysis should be performed (i.e. other methods to detect virus infectivity and pathological effect. My main concerns are that:

Currently, the efficacy of commercial vaccines, mainly, PCV2a-based vaccines, is still under debate due to the simultaneous circulation of many PCV2 variants in domestic farms. From an experimental point of view, the evaluation of vaccine efficacy in farms is challenging because, unlike controlled laboratory experiments, there are several uncontrollable factors at play. These factors include the complex pattern of virus variants, the duration and intensity of wild virus exposure to pigs at different stages of rearing, the level of maternal antibodies during wild virus infection, the immune status of pigs exposed to the wild virus, and the presence of co-infection with other pathogens. These unpredictable and uncontrollable factors are therefore daily challenges in real pig farms. For this reason, we conducted a trial to assess the efficacy of a new vaccine in controlling prevalent variants. This trial took place on three different farms with different conditions, such as farm size, vaccination program, herd healthy status, and management protocols. In this trial, the control group can be viewed as a challenge group exposed to wild virus in unvaccinated pigs at some stage of rearing, analogous to the control group with post-challenge in a controlled laboratory experiment. Similarly, the results from the vaccinated group in the farm trial were expected to be similar those of the vaccinated group in the laboratory experiment, demonstrating the protective effect of the vaccine following challenge with wild virus after vaccination. Therefore, these trial conditions will be highly beneficial for validating vaccine efficacy in real farm-specific scenarios, involving exposure to wild virus at unspecified time points. This clinical trial approach is in marked contrast to the laboratory-based efficacy testing conditions. In addition, this type of validation of the vaccine efficacy can be uniquely performed in a farm scenario rather than a laboratory setting.  In conclusion, these results are of greater value to pig farmers and practitioners as they provide insights into the expected results of the new vaccine use under authentic farm conditions, influenced by various factors. Therefore, the aim of this trial was to provide practical, rather than purely scientific, data that would ultimately shed light on why the vaccine should contain the strain currently circulating on pig farms. We described in Conclusions section (line401-406).

 1- It seems that most of the responses detected are due to the field infection and the effect of vaccination is minimal.

Assessing the efficacy of a circovirus vaccine in on-farm trials is challenging because single circovirus infection of piglets usually does not result in obvious clinical signs. Consequently, the efficacy of PCV2 vaccine is usually determined by two measures: significant differences in the occurrence or concentration of viremia after diminishing maternal antibodies, and varying ADWG over the same period. Thus, vaccine success is indicated by complete virus elimination or viremia reduction, together with a significant increase in ADWG of vaccinated pigs compared to unvaccinated pigs, as shown in these results.  In this study, histopathological analysis showed no significant difference in lesions between the two groups (data no shown). This is likely to be due to insignificant histological variation caused solely by circulating circovirus and ongoing non-clinical effects of other pathogens on farms. With regard to circulating antibodies, the typical pattern of antibody levels is common in virus-free farms for both vaccinated and non-vaccinated groups. However, in farms such as the trial farms where different circoviruses are circulating, assessing vaccine efficacy by ELISA alone or changes in neutralizing antibodies is complex. Notably, a clear pattern emerges in the scenarios: antibody changes reverse during the rearing phase, indicating suppression of viremia by vaccine-induced antibodies. Such antibody reversal pattern have been observed in numerous reports, supporting our findings from these farm trials.  In the case of viremia, a significant difference was shown between vaccinated and non-vaccinated groups in the fattening stage of pigs, as a result of suppression of early infection with circovirus in young piglets (in Figure 4) In summary, given the reversal of antibody levels, the reduction in viremia and the increase in ADWG, we are confident that the efficacy of the vaccine has been convincingly demonstrated at the farm level. We described in Discussion section (line373-395).

   2-The different farms received different vaccination regimes.

The farms used in this trial were typical pig production units, each following their own vaccination schedule and management protocols. The aim of this trial was to evaluate the efficacy of the new vaccine under existing farm conditions, without any changes. As a result, the trial was conducted in three different farm environments, where the vaccination regimes are expected to vary.

 3- The authors have not tested for maternal antibodies which may affect the results.

The level of circulating antibody prior to the start of this trial can be used as an indicator of the level of maternal antibody in the piglets transferred from the sow. The antibody level of the 3-week-old piglets was assessed by blood sampling just prior to vaccination and is labeled 0 wpv in the figures. The reduced antibody levels in the unvaccinated control group at 4 wpv can be attributed to the eventual decline of maternally transferred antibodies, as the piglets were vaccinated shortly after the 3-week-old piglets were sampled. Conversely, an increase in vaccine-induced antibody levels was observed in the vaccinated group.  The antibody levels at 3 weeks of age (maternal antibody) in test piglets, selected from the three farms, and the pattern of variation in neutralizing antibodies caused by wild viruses on the farms are shown in Figure 3.

 4- The authors used PCR as an indication of infection instead of the virus isolation. Neither the sequence of the primers used nor the target product was provided which important to evaluate the study.

Revised in Phylogenetic analysis of Materials and Methods(line147-152).

 Some typos and wrong phrases were use

We get English Editing service again.

Thank you.
Sincerely yours,

Seok-Jin Kang

Reviewer 3 Report

This is an interesting papers showing the results of clinical trial of pigs vaccinated with viral-like particle. vaccine against PCV2d variant currently circulating in a number of Asian countries. The papers seems to be improved since its last conditions. In view of my expertise it may be accepted in its current form.

Round 2

Reviewer 1 Report

Thank you for revising this paper-I am pleased to see the ARRIVE guidelines being followed with more information on interventions, monitoring, animal husbandry etc 

I think it would be useful to have more descriptive statistics in the abstract, analyses on effect sizes in addition to significance and also better descriptions of the data in your figure legends. Figure legends should stand alone, details of what your error bars illustrate and pertinent information on statistical relevance.

Author Response

Thank you very much for giving us a chance to revise our manuscript entitled ‘A Field efficacy Trial of Recombinant Porcine Circovirus Type 2d Vaccine in Three Herds’ by Ju et al. All comments made by the reviewers were quite valid and we sincerely appreciate for reviewers’ endeavor to improve the quality of our manuscript. We considered all suggestions of the reviewers for revising the original version of manuscript. We prepared a point-by-point report to explain our revision in details and also annexed a highlighted copy of the revised manuscript. However, we are still afraid of remaining of any mistake, and misunderstanding and neglecting of reviewers’ comments. We of course are ready for further revising our manuscript as the reviewers’ request. 

Thank you for revising this paper.I am pleased to see the ARRIVE guidelines being followed with more information on interventions, monitoring, animal husbandry etc I think it would be useful to have more descriptive statistics in the abstract, analyses on effect sizes in addition to significance and also better descriptions of the data in your figure legends. Figure legends should stand alone, details of what your error bars illustrate and pertinent information on statistical relevance

Revised as reviewer’s comment

Thank you.
Sincerely yours,

Seok-Jin Kang

Reviewer 2 Report

I appreciate the authors response but I still think that the design of the study is poor and there is a little effect of the vaccines if any. My suggestion is to try to turn study into studying the immune responses induced in different groups and try to find some predictors of virus replication or correlation between antibody response and other health parameter.

Language have been improved but there are still some errors here and there

Author Response

Thank you very much for giving us a chance to revise our manuscript entitled ‘A Field efficacy Trial of Recombinant Porcine Circovirus Type 2d Vaccine in Three Herds’ by Ju et al. All comments made by the reviewers were quite valid and we sincerely appreciate for reviewers’ endeavor to improve the quality of our manuscript. We considered all suggestions of the reviewers for revising the original version of manuscript. We prepared a point-by-point report to explain our revision in details and also annexed a highlighted copy of the revised manuscript. However, we are still afraid of remaining of any mistake, and misunderstanding and neglecting of reviewers’ comments. We of course are ready for further revising our manuscript as the reviewers’ request. 

I appreciate the authors response but I still think that the design of the study is poor and there is a little effect of the vaccines if any. My suggestion is to try to turn study into studying the immune responses induced in different groups and try to find some predictors of virus replication or correlation between antibody response and other health parameter.

This study is the first outdoor application trial of the PCV2d VLP vaccine, which has previously demonstrated efficacy against PCV2d challenge in preclinical studies [Kang et al., 2021 (pathogens) & 2023 (vaccines, in press)]. The two previous studies are about the development process and efficacy evaluation of the PCV2d vaccine strain used in this study. Therefore, the aim of this field trial was to evaluate the actual protective effect of the vaccine when used in field conditions, providing valuable insights for farmers and veterinarians who utilize the vaccine. The study design was based on “the guidelines for outdoor clinical trials of animal biological products”, ensuring a comprehensive evaluation of both efficacy and safety. The evaluation focused on assessing the vaccine's ability to protect against natural PCV2 infection within these farm settings. Although evaluating the efficacy of the vaccine in the presence of various environmental factors is challenging in field, we observed induction of immune response, reduction in viremia and an increase in body weight, representative indicators of PCVAD, in the vaccine group. Recently, there has been a rapid increase in the global prevalence of PCV2d genotype. Furthermore, there has been an increase of vaccine failure cases associated with the currently available commercial vaccines based on the PCV2a genotype. Therefore, there is a growing demand for vaccines targeting the prevailing PCV2d genotype. Since 2019, we have been developing a PCV2d vaccine using PCV2d. As a result, PCV2d vaccine product was approved through the preclinical study as well as field trials of this study. These findings provide meaningful results that can be shared with industry professionals, producers, and veterinarians. We would greatly appreciate it if you could thoroughly consider the aspects we have mentioned. The reviewer’s suggestion is a valuable idea, and we are actively considering and working on it. We are also eager to explore the topics you mentioned. Actually, PCV2 vaccine is very difficult to be evaluated because lack of distinct indicators against a wasting disease. Therefore, it needs additional indicators of vaccine efficacy evaluation beyond increase of body weight and reduction of viremia. As the reviewer’s suggestion, we should find some predictors or indicators and try to do in the future study.

 Comments on the Quality of English Language:Language have been improved but there are still some errors here and there

We will proceed with additional English editing service.

This field trial was the first evaluation of PCV2d commercial vaccine

Thank you.
Sincerely yours,

Seok-Jin Kang